# Comparison of Helical Blade Systems for Osteoporotic Intertrochanteric Fractures Using Biomechanical Analysis and Clinical Assessments

**DOI:** 10.3390/medicina58121699

**Published:** 2022-11-22

**Authors:** Hyeonjoon Lee, Sang Hong Lee, Wonbong Lim, Seongmin Jo, Suenghwan Jo

**Affiliations:** 1Department of Orthopaedic Surgery, Chosun University Hospital, Gwangju 61453, Republic of Korea; 2School of Medicine, Chosun University, Gwangju 61452, Republic of Korea

**Keywords:** biomechanics, cut-out, cut-through, helical blade, intertrochanteric fracture

## Abstract

*Background and Objectives*: This study aimed to compare the biomechanical properties and outcomes of osteoporotic intertrochanteric fractures treated with two different helical blade systems, the trochanteric fixation nail-advanced (TFNA) and proximal femoral nail antirotation II (PFNA), to evaluate the efficacy and safety of the newly introduced TFNA system. *Materials and Methods*: A biomechanical comparison of the two helical blades was performed using uniaxial compression tests on polyurethane foam blocks of different densities. The peak resistance (PR) and accumulated resistance (AR) were measured during the 20 mm advancement through the test block. For clinical comparison, 63 osteoporotic intertrochanteric fractures treated with TFNA were identified and compared with the same number of fractures treated with PFNA using propensity score matching. Ambulatory status, medial migration, lateral sliding, fixation failure, and patient-reported outcomes were compared between the two groups over a minimum of 1 year’s follow up. *Results*: The uniaxial compression test showed that a slightly, but significantly lower resistance was required to advance the TFNA through the test block compared with the PFNA (20 PCF, *p* = 0.017 and *p* = 0.026; 30 PCF, *p* = 0.007 and *p* = 0.001 for PR and AR, respectively). Clinically, the two groups showed no significant differences in post-operative ambulatory status and patient-reported outcomes. However, in TFNA groups, significantly more medial migration (TFNA, 0.75 mm; PFNA, 0.40 mm; *p* = 0.0028) and also, lateral sliding was noted (TFNA, 3.99 mm; PFNA, 1.80 mm; *p* = 0.004). Surgical failure occurred in four and two fractures treated with the TFNA and PFNA, respectively. *Conclusions:* The results of our study suggest that the newly introduced TFNA provides clinical outcomes comparable with those of the PFNA. However, inferior resistance to medial migration in the TFNA raises concerns regarding potential fixation failures.

## 1. Introduction

Intertrochanteric fractures are recognized as one of the most common and devastating fractures in elderly patients with osteoporosis [1]. Early rehabilitation is recommended to optimize outcomes; therefore, prompt fixation using rigid implants is paramount. A number of internal fixation methods have been suggested, and intramedullary nails are currently recognized as the treatment of choice, providing satisfactory outcomes [2].

One of the most commonly used implants for intertrochanteric fracture fixation is the proximal femoral nail with a helical blade. Currently, two types of helical blade systems from the same manufacturer are widely used: the trochanteric fixation nail-advanced (TFNA, Depuy Synthes, West Chester, PA, USA) and proximal femoral nail antirotation II (PFNA, Depuy Synthes, West Chester, PA, USA). Both helical blade systems were designed to provide more bone purchase on the femoral head, thereby providing better stability [3,4]. However, while the safety and efficacy of the PFNA system have been validated in numerous studies, limited clinical data are available on the newly introduced TFNA system. The helical blades in the two nail systems have different morphological features, and whether a single proximal nail system provides better stability is unclear. Theoretically, a larger core diameter may provide better purchase on the trabecular bone within the femoral head; therefore, the smaller core diameter in the TFNA system (10.35 mm) may result in less stability [5]. However, this hypothesis has not been previously validated.

The aim of the current study is to compare the biomechanical properties and the clinical outcomes of osteoporotic intertrochanteric fracture treatment with the TFNA with those of the PFNA to validate the safety and efficacy of the newly introduced TFNA system.

## 2. Materials and Methods

### 2.1. Biomechanical Comparison of Resistance to Axial Compression—A Pilot Study in Laboratory Simulated Model

A biomechanical test was performed to assess resistance to migration. To achieve this, testing blocks (rigid polyurethane foam blocks, Sawbones, USA) were prepared to measure the axial compression stress of the two helical blades. Two types of test blocks of different densities (20 PCF and 30 PCF) were utilized to simulate bones of different densities. According to the manufacturer, the test blocks have compressive moduli of 210 MPa and 445 MPa and tensile moduli of 284 MPa and 445 MPa for PCFs of 20 and 30, respectively. Axial compression tests were performed on the helical blades of the TFNA and PFNA systems, both 85 mm in length, using the test blocks. For the initial setup, the helical blade was anchored to a servo-hydraulic universal test machine (MTS, Bionix Landmark 370, MTS Systems Corporation, USA). This was then advanced until the tip of the screw contacted the test block (Figure 1). The helical blade was further advanced axially through the test block at 15 mm/min to a depth of 20 mm. The load–displacement curve was acquired during the course of the advancement, and the resistance to migration was recorded to compare the two helical blades. To facilitate the analysis, noisy data were smoothed using MATLAB (The MathWorks, Natick, MA, USA). Resistance to migration is defined as (1) peak resistance and (2) accumulated resistance. The peak resistance was defined as the maximum load reached during the advancement of the helical blade, while the accumulated resistance was defined as the area under the load–displacement curve, and the test was repeated seven times for each type of helical blade [6].

### 2.2. Clinical Comparison

#### 2.2.1. Patient Selection

One-hundred and one consecutive patients who underwent TFNA fixation between January 2019 and June 2020 in our institution were identified. Of this cohort, three patients were excluded because the fractures were caused by high-energy trauma, and one patient was excluded for fixation using a screw-type lag screw. An additional 11 patients were excluded for fractures extending to the subtrochanteric region, which required a long nail, and 23 patients were excluded due to death or unavailability for follow-up. The remaining 63 patients constituted the basis of our study. The same number of patients were selected from our database of 412 patients who underwent fixation with the PFNA system for osteoporotic intertrochanteric fractures between May 2013 and May 2021. By applying the same exclusion criteria to this cohort group, 145 patients were excluded, leaving 267 patients. Patient selection was based on propensity score matching with age, sex, body mass index (BMI), American Society of Anesthesiologists (ASA) classification, fracture classification according to the AO/OTA classification, and quality of reduction (Figure 2). Table 1 shows the epidemiologic characteristics of the enrolled patients, including age, sex, bone marrow density (BMD), body mass index (BMI), American Society of Anesthesiologists (ASA) classification, and AO/OTA fracture classification [7].

#### 2.2.2. Description of Surgical Technique, Post-Operative Management and Assessments

All surgeries in both groups were performed on a regular surgery table by a single surgeon. Initially, closed reduction with fluoroscopic guidance was attempted, and if the reduction was unsatisfactory, percutaneous aid with a K-wire or sharp Hohmann retractor was utilized. The aim was to achieve anatomical approximation. If not possible, extramedullary reduction was attempted with the lag screw positioned at the medial or slightly inferior portion of the femoral head and the tip-apex distance within 25 mm. Following insertion of the helical blade, interfragmentary compression was achieved by manually turning the compression nut. The TFNA system has additional rotational locking for a controlled collapse and compression. In all cases, short-length nails (170, 200, or 235 mm) were utilized with a helical blade. Postoperatively, all patients received thromboprophylaxis and intermittent pneumatic compression devices to prevent deep vein thrombosis [8]. The use of a wheelchair was allowed on the next day or whenever the patient was comfortable using a wheelchair. Ambulation was allowed as tolerated by the patients. All patients were encouraged to walk using an assistive device for two weeks following surgery. 

Immediate postoperative radiographic assessment was performed by measuring the tip-apex distance, position of the helical blade, and reduction quality using simple radiography. The tip-apex distance was defined as the sum of the distance from the tip of the lag screw to the apex of the femoral head on anteroposterior and lateral radiographs [9,10]. The position of the helical blade was classified using the Cleveland index, and the fracture reduction quality was assessed using the Baumgartner classification as good, acceptable, or poor [11]. Postoperatively, two orthopedic surgeons evaluated the radiographs. In case of disagreement, senior orthopedic surgeons were consulted, and the interobserver reliability between the first two observers was evaluated.

The patients were followed up at regular intervals for a minimum of 1 year. Bone union status and the bone-implant relationship were monitored. At the last follow-up, patients were evaluated using the modified Harris hip score, Koval grade, and presence of lateral thigh pain. The radiographic assessment at this time was aimed at determining medial migration and lateral sliding of the helical blade, as well as failure of fixation or the implant.

### 2.3. Statistical Analysis

SPSS 27.0 (SPSS Inc., Chicago, IL, USA) was used for propensity score matching and the analysis of the results after the biomechanical experiment and clinical outcomes [12]. The variables are presented as a means with standard deviations. Interobserver reliability was evaluated using Cohen’s kappa. The criterion for acceptable reliability was a kappa value ≥ 0.60. For the biomechanical test, the resistance from uniaxial compression between the two angle blade systems was compared using the Mann–Whitney test. For clinical comparison, Fisher’s exact test or the Chi square test was used to compare categorical data, while the Student’s *t*-test was used to compare continuous variables. The level of significance was set at *p* < 0.05. 

## 3. Results

### 3.1. Biomechanical Comparison

A slightly but significantly lower peak resistance was observed in the helical blade of the TFNA system, which was 3.1% and 2.3% smaller in the PCF 20 and PCF 30 test blocks, respectively (*p* = 0.017 and 0.007, respectively). The accumulated resistance for 20 mm migration was also significantly lower in the TFNA system, which was 9.7% and 13.6%, respectively, in the PCFs 20 and PCFs 30 test blocks (*p* = 0.026 and 0.001, respectively) (Figure 3).

### 3.2. Clinical Comparison

Prematched and postmatched demographic characteristics of the TFNA and PFNA groups are shown in Figure 4 and Table 1. The demographic characteristics did not differ significantly between the TFNA and PFNA groups.

Bone union was achieved in 15% and 13% of patients in the TFNA and PFNA groups, respectively, at 3 months and in 65% and 61% of patients in the TFNA and PFNA groups, respectively, at 6 months. Overall union was achieved in 92% of the patients treated with TFNA and 89% of the patients treated with PFNA at 1-year follow-up.

Radiographic and clinical outcomes are presented in Table 2. Cohen’s kappa value was 0.631 (*p* < 0.00), showing overall substantial agreement for the assessment of fracture reduction status. The mean medial migration in the PFNA group was 0.41 mm and that in the TFNA group was 0.75 mm, reaching statistical significance (*p* = 0.0028). The mean lateral migration in the PFNA group was 1.80 mm and that in the TFNA group was 3.99 mm, reaching statistical significance (*p* = 0.004).

Fixation failure occurred in two cases in the PFNA group and four cases in the TFNA group. The epidemiology, failure mechanism, and subsequent surgery for fixation failure are listed in Table 3. Of the six patients who developed fixation failure, four underwent subsequent arthroplasty, while two patients did not undergo additional surgery due to medical insufficiency. 

## 4. Discussion

The results of the present study indicate that the TFNA system provides clinical outcomes comparable to those of the PFNA system. However, the small core diameter of the TFNA may provide inferior biomechanical stability, which may lead to failure by the cut-through phenomenon.

The helical blade system has been reported to provide better varus and rotational stability than conventional lag screws; however, excessive medial migration remains a significant problem [3]. The clinical outcomes of treatment with the PFNA system are satisfactory, but those of treatment with the TFNA system are lacking, as it reached the global market only in 2015 [13,14]. In addition, the first use of this implant in our region was in 2018; therefore, limited data are available on its use in the Asian population [15].

Both the TFNA and PFNA systems are provided by the same manufacturer with a similar design concept of providing better fixation stability in osteoporotic fractures by impacting the bone on the helical blade. The TFNA was developed to improve its precursor, the trochanteric fixation nail (TFN) system (Depuy Synthes, West Chester, PA, USA), by introducing a lateral relief cut to reduce impingement on the lateral cortex and a bump cut on the mid-portion of the proximal aperture to improve fatigue strength [2,16,17,18]. Despite its proposed advantages, a recent study by Lambers et al. reported 16 cases of TFNA implant breakage, raising concerns about its safety [19]. Such a failure did not occur in our series; however, our study consisted of only 63 patients in each group. We theorize that implant failure might have been recorded in a larger sample size.

The current study focuses on the design of a helical blade in a two-nail system. The helical blade in the TFNA provides a cement augmentation option that allows bone cement to penetrate through small holes in the helical blade onto the femoral head to increase the load to failure [20]. However, the cement option is not yet available in our region, and without cement, the core diameter may have a significant effect on stability. According to the specifications, the helical blade of the PFNA has a core diameter of 12.2 mm and that of the TFNA is 10.35 mm. In addition, the PFNA consists of four flanges while the TFNA has three. Consequently, it can be hypothesized that the TFNA system may be less stable in terms of varus and axial migration [21,22]. In accordance with our hypothesis, we found the PFNA to provide more resistance to axial compression, which is similar to medial migration in a clinical setting. However, the difference was less than 10%, and its clinical significance remains unclear. Clinically, we found comparable bone healing capacity and outcomes using both nail systems. However, we also found significantly more medial migration in the TFNA system, suggesting that the helical blade in the TFNA system may provide less resistance to cut-through. This is consistent with the greater number of failures occurring in TFNA use. 

Our study has several limitations. First, the study sample may have been too small and the study was designed in a retrospective fashion. Second, the operating surgeon had performed a greater number of fixations using the PFNA system, and familiarity with the PFNA system may have affected the results. Third, the osteoporotic medication administered was not taken into account. We tried to minimize the bias by matching the two groups with propensity scores; however, we believe that a prospective study with a greater number of patients may provide more accurate results. Moreover, we focused on the difference in helical blade design, but other morphologic differences, such as the proximal diameter or curvature of the nail, may also have significantly affected the outcome [23]. However, existing studies suggest that such morphological differences between the two nails do not influence clinical outcomes [15,18]. Finally, from a biomechanical perspective, our protocol tested axial loading of the lag screw on the foam bone, which is not the direction in which the load is applied in vivo.

## 5. Conclusions

This is the first study to compare the outcomes of intertrochanteric fractures treated with two commonly used helical blade systems, and the results of our study suggest that the newly introduced helical blade system (TFNA) provides comparable clinical outcomes as compared to PFNA. However, inferior resistance to axial loading as demonstrated in biomechanical tests and the evidence toward more medial migration in radiographic follow-up raises concern for potential cut-through failure with the use of the TFNA system. 

## Figures and Tables

**Figure 1 medicina-58-01699-f001:**
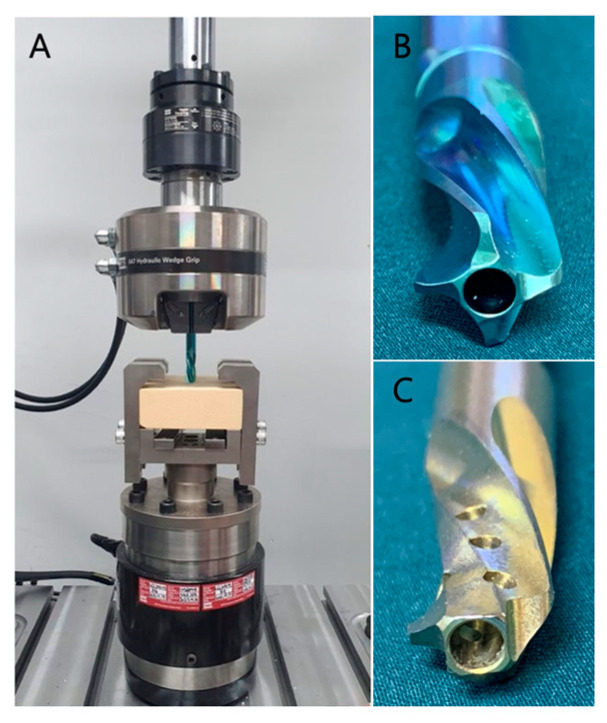
(**A**) Test set-up to analyze helical screw system resistance to migration. Helical blade in (**B**) PFNA and (**C**) TFNA system. Note the difference in core diameter and number of flanges between PFNA and TFNA.

**Figure 2 medicina-58-01699-f002:**
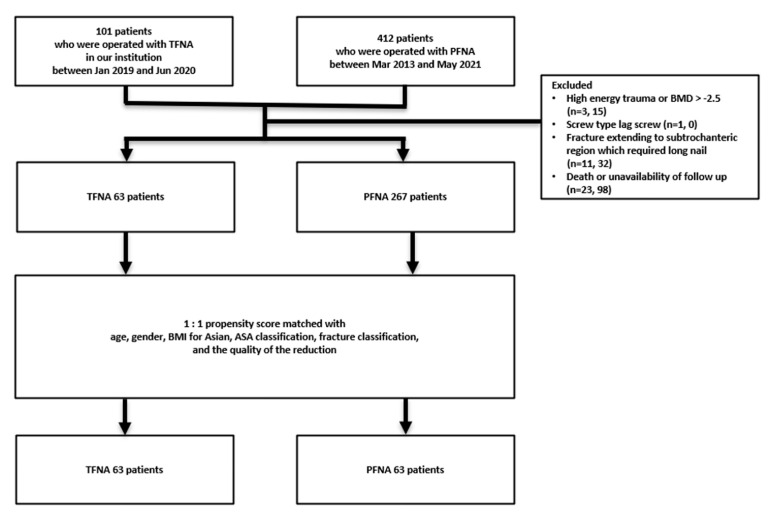
Flowchart with a summary of patient enrollment and propensity score matching.

**Figure 3 medicina-58-01699-f003:**
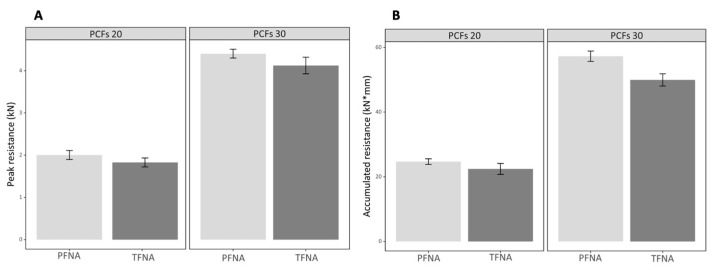
Peak resistance (**A**) and accumulated resistance (**B**) of the two helical blades on test blocks of different densities (PCFs 20 and PCFs 30).

**Figure 4 medicina-58-01699-f004:**
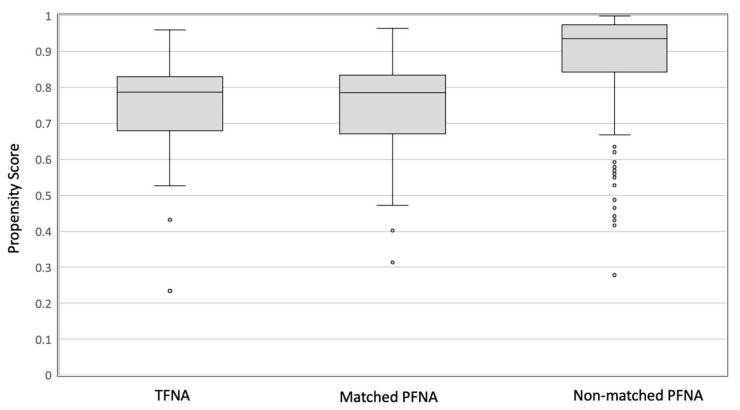
Box and whisker plot of the logit propensity scores of patients with osteoporotic intertrochanteric fractures treated with TFNA and PFNA systems.

**Table 1 medicina-58-01699-t001:** Demographic and baseline characteristics of patients with osteoporotic intertrochanteric fractures treated with the TFNA and PFNA systems.

	TFNA (*n* = 63)	PFNA (*n* = 63)	*p*-Value
Mean age (years)	79 ± 9.3	80 ± 8.9	0.575
Sex (% male)	27.2%	29.3%	0.812
BMD (T-score)	−3.47 ± 1.2	−3.62 ± 1.5	0.560
BMI (kg/m^2^)	22.75 ± 7.1	21.08 ± 7.5	0.387
Ambulatory status prior to fracture (Koval grade)	2.4 ± 1.29	2.2 ± 1.55	0.411
ASA score			0.936
ASA 1	1.6%	3.2%	
ASA 2	17.5%	22.2%	
ASA 3	65.1%	74.6%	
AO classification			0.051
A1	20.6%	28.6%	
A2	50.1%	46.0%	
A3	28.6%	25.4%	
TAD (mm)	19.9 (±4.9)	18.5 (±5.6)	0.135
Position of lag screw in Cleveland index			0.119
1, 2, 3, 4, 7	2	4	
5, 6, 8, 9	61	59	
Fracture reduction status			0.870
Good	68.3%	71.4%	
Acceptable	27.0%	25.4%	
Poor	4.8%	3.2%	

**Table 2 medicina-58-01699-t002:** Comparison of clinical outcomes of patients with osteoporotic intertrochanteric fractures treated with the TFNA and PFNA systems.

	TFNA (*n* = 63)	PFNA (*n* = 63)	*p*-Value
Follow up duration (mo)	19.6 ± 12.8	24.1 ± 8.89	0.048 *
Bone union			0.041 *
Within 3 months	15%	13%	
Within 6 months	65%	61%	
At 1 year	92%	89%	
Medial migration (mm)	0.75 ± 2.4	0.41 ± 1.8	0.003 *
Lateral sliding (mm)	3.99	1.80	0.004 *
Modified Harris hip score	76.6 ± 16.3	74.3 ± 14.3	0.162
Koval grade	2.1 ± 0.3	2.3 ± 0.8	0.454

(* *p* < 0.05).

**Table 3 medicina-58-01699-t003:** Characteristics of fixation failure in osteoporotic intertrochanteric fractures treated with the TFNA and PFNA systems.

Age/Sex	Treatment Type	Blade Position (AP/Lat.)	Initial TAD (mm)	Failure Mechanism	Subsequent Operation
81/F	TFNA	Center/Center	24.4	Cut-out	Arthroplasty
74/F	TFNA	Sup/Center	20.1	Varus collapse	None
71/F	TFNA	Center/Center	18.9	Cut through	Arthroplasty
84/F	TFNA	Center/inf.	4.5	Cut through	None
86/F	PFNA	Center/inf.	18.8	Cut-out	Arthroplasty
75/M	PFNA	Sup./Sup.	14.4	Implant breakage	Arthroplasty

## Data Availability

The dataset used and analyzed during the current study are available from the corresponding author on a reasonable request.

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
