# Peer review of "Comparison of Helical Blade Systems for Osteoporotic Intertrochanteric Fractures Using Biomechanical Analysis and Clinical Assessments"

_medicina, 2022, doi:10.3390/medicina58121699_

Round 1

Reviewer 1 Report

In the discussion section, you could cite article

Yuan H, Acklin Y, Varga P, Gueorguiev B, Windolf M, Epari D, Schuetz M, Schmutz B. A cadaveric biomechanical study comparing the ease of femoral nail insertion: 1.0- vs 1.5-m bow designs. Arch Orthop Trauma Surg. 2017 May;137(5):663-671. doi: 10.1007/s00402-017-2681-3. Epub 2017 Apr 3. PMID: 28374092.

Author Response

Thank you for hiring the valuable time to review our manuscript and we greatly appreciate the excellent review. 

We are aware of the study recommended by the reviewer (Yuan et al, 2017) and we agree that there are numerous factors, other than helical blade shape, that may influence the outcome. Therefore, all authors agreed that adding this to the discussion session will strengthen our manuscript and we added the statement with the references. We thank the author for the valuable advice.

The change in the manuscript is as follows : 

Moreover, we focused on the difference in helical blade design, but other morphologic differences, such as the proximal diameter or curvature of the nail, may also have significantly affected the outcome. [23] (Yuan et al, 2017)

In addition, we have also included references and content in the introduction section so that sufficient background is provided. 

Reviewer 2 Report

General comments

I understood the authors would like to describe TFNA provides comparable clinical outcomes with those of the PFNA., however, TFNA raises concerns regarding potential fixation failures. Their findings were not incorrect, however, similar to previous studies, such as Peter et al. Acta Orthop less 2021 ect, less informative other than in terms of reporting on Asians. Moreover, I think their conclusion seemed to be unclear. TFNA should not be used any more ? Is there good or poor indication of TFNA ? I hope the authors should highlight the novel findings from their results.

Specific comments

- I could not understand Figure 4. What does the horizontal axis mean? The authors should be clarified.

Author Response

First of all, we thank the reviewer for hiring the valuable time to review our manuscript and we greatly appreciate the important comments and reviews. 

Following the comments and reviews, we have changed the manuscript as follows :

1. We are aware of the study by Peter Schmitz et al and agree that our manuscript is limited by the small number of patients cohort. However, while the paper from Peter Schmitz et al compared the result of the conventional lag screw (gamma short nail) vs helical blade (TFNA), our manuscript is a comparison between two different types of helical blades (PFNA and TFNA) which we believe has not been done previously. We initiated the study because the two helical blades have different morphological shapes yet how this difference affects the biomechanics or clinical outcome is not described in any literature or even by the manufacturers. Also, as our study is limited in number, we tried to compensate for this by comparing the two groups using propensity scores. 

2. We agree with the reviewer's comment on the conclusion and we corrected the conclusion as follows: This is the first study to compare the outcomes of intertrochanteric fractures treated with two commonly used helical blade systems, and the results of our study suggest that the newly introduced helical blade system (TFNA) provides comparable clinical outcomes as compared to PFNA. However, inferior resistance to axial loading as demonstrated in biomechanical tests and the evidence toward more medial migration in radiographic follow-up raises concern for potential cut-through failure with the use of TFNA. 

From our result, we can not make a firm conclusion on whether TFNA should not be used. Although we had a comparable clinical result, we also have a concern with the TFNA's inferior resistance to medial migration and therefore, if we only consider the migration of the helical blade, the TFNA do not seem to provide better result than PFNA if it is used without cement augmentation. However, there are other features of TFNA, such as the range of curvature, that seem to provide an advantage which is not discussed in the current study. Therefore, we added this as the limitation of the current study. 

3. We apologize for not providing sufficient information in figure 4. Figure 4 compares the propensity score among the patients with TFNA, the matched patients with PFNA, and the non-matched patients with PFNA to show that the patients were properly matched for comparison. We have corrected the figure and the legend.

Thank you again for the excellent reviews and we were able to strengthen our manuscript because of the reviewer's comment. However, we humbly asked the reviewer to consider that despite the limitations of the manuscript, this is the first study that compares the PFNA and TFNA which was done both biomechanically and clinically. 

Reviewer 3 Report

Dear Authors 

My comments on your paper 

-Well written study 

- my only concern is the methods adapted in materials and methods /// that is if this lab set up can be mimicked in real life situation to be able to comment on the outcomes . 

- so it can be just reported as a pilot study in lab simulated models  in the heading so that the reader will be well aware about the study 

- the readers will be benefitted if photographs of both implants are given

Thank You 

Author Response

Thank you for hiring the valuable time to review our manuscript and we greatly appreciate the excellent review. 

We agree with the reviewer's concern that our lab setup may not properly mimic the real-life situation. Therefore, the following sentence was added as a limitation of the current study: from a biomechanical perspective, our protocol tested axial loading of the lag screw on the foam bone, which is not the direction in which the load is applied in vivo.

Also, following the reviewer's recommendation, the section heading was corrected as follows : Biomechanical Comparison of Resistance to Axial Compression – A pilot study in Laboratory Simulated Model

We have also added a picture of the helical blade of the TFNA and PFNA (Figure 1b, 1C) so that that the readers would have a better understanding. Also, the morphological differences between the two helical blades are described in detail in the discussion section. 

We thank the reviewer again for making important comments to improve our manuscript.

Round 2

Reviewer 2 Report

I think the authors made appropriate revisions for publication. Sincerely, Norio Imai